# On the use of kinship and familiarity associated social information in mediating *Drosophila melanogaster* oviposition decisions

**Emily Rakosy**[1,2]*, **Sanduni Talagala**[1], **Tristan A. F. Long**[1]

**1** Department of Biology, Wilfrid Laurier University, Waterloo, Ontario, Canada, **2** Department of Cell & Systems Biology, University of Toronto Mississauga, Ontario, Canada

* emily.rakosy@alumni.utoronto.ca

## Abstract

Decisions where an individual lays their eggs are important, as the choice may affect their offspring's survival and lifetime reproductive success. Information produced by conspecifics can potentially be useful in making decisions as this "social information" may provide an energetically cheaper means of assessing oviposition site suitability rather than acquiring it personally. However, as not all public information may be equally beneficial, cues produced by kin may be especially valuable as they might signal suitable microenvironments, and are associated with other fitness advantages resulting from improved foraging success and/or a decreased risk of competition/cannibalism compared to sites where unrelated conspecifics are located. Using the fruit fly, *Drosophila melanogaster*, we explored whether public information use is associated with kin-based egg-laying decisions. Kinship is potentially recognized in several ways, including environmentally-associated proxy cues, so we explored whether there were biases in how focal females interacted with cues from conspecifics that differed in both genetic relatedness, and environmental "familiarity." In a series of inter-connected assays, we examined the behaviour of focal females that interacted with a choice of potential egg-laying substrates that differed in the manner of their prior conspecific exposure, and counted the offspring that eclosed from these different substrates. Sites that had exhibited cues produced by conspecific demonstrators were visited more, and yielded more focal offspring compared to unexposed substrates. Furthermore, patterns of bias in offspring production were consistent with ovipositing females exhibiting sensitivity to the kinship status of the prior substrate's occupants. The basis of the kinship categorization by ovipositing females appears to be based on phenotypes that reflect true genetic relatedness, but the nature of the social information can be affected by other factors. These results further highlight the potential usefulness of *D. melanogaster* as a model to understand the evolution of social behaviour in the expression of decision-making.

**Data availability statement:** All data analysed in this manuscript are archived in Borealis, the Canadian 47 Dataverse Repository: https://doi.org/10.5683/SP3/N7XBOJ.

**Funding:** TAFL received support in the for of Natural Sciences and Engineering Research Council (https://www.nserc-crsng.gc.ca) Discovery grants (RGPIN-2016-06133 and RGPIN-2022-03988). The funding agency played no role in the study design, data collection and analysis, decision to publish, or preparation of the manuscript.

**Competing interests:** The authors have declared that no competing interests exist.

## Introduction

In species where there is little-to-no post-natal parental care, decisions about where and when an individual should lay their eggs are likely very important, as there are potentially dire fitness outcomes for one's offspring if unsuitable oviposition sites are chosen [1,2,3]. In order to make better decisions, using cues produced by conspecifics (aka "social information" *sensu* [4]) may be of great value to the individual as they may provide meaningful information about potential egg-laying sites that can be obtained at a lower energetic cost than having to explore and evaluate alternate options oneself. Social information can be valuable in reducing an individual's uncertainty about their environment, and may help them locate areas that are rich in important resources such as food, shelter, desirable mates, increased foraging success, or are at a lower risk of predation/parasitism, any of which may benefit the mimicking individual and/or their offspring [5,6,7]. Furthermore, the energy savings of using social information over private information can also be redirected into increased provisioning and/or production of offspring, thereby enhancing the benefits of its use in making decisions [8].

While fruit flies, *Drosophila melanogaster*, have a long and storied use as a model in studies of genetics, development, behaviour and evolution [9,10], they are also emerging as a powerful model in which to study social learning and the use of social information [11,12]. *D. melanogaster* has been found to use the visual, gustatory, tactile, and olfactory cues produced by conspecifics in several different decision-making contexts. For instance, flies use social cues when engaged in spatial learning tasks to improve the speed that a "safe zone" is detected [13]. Virgin female flies that are exposed to deposits left by conspecific demonstrators exhibit lower post-mating egg-laying rates than unexposed females, presumably because they are better able to anticipate the forthcoming availability of oviposition sites and the potential degree of future larval competition [14]. Mate choice can involve social information use as female fruit flies are more likely to mate with males exhibiting similar phenotypic characteristics as a male they've observed successfully mating with another female [15–18]. While the robustness of this phenomenon is debatable [19] it may still be advantageous because mate choice is often costly to females [20], and copying the decisions made by conspecifics represents an economical adaptive strategy [16]. Important information about environmental conditions can also be obtained indirectly by flies interacting with conspecifics. In a study by Kacsoh et al. [21], female flies decreased their oviposition rate in response to exposure to parasitic wasps, and this defensive change in egg-laying behaviour was adopted by inexperienced flies housed with the wasp-exposed females, with the cues about environmental risk communicated visually using the demonstrator flies' wings. Thus, while *D. melanogaster* has not historically been considered a "social" species [12], it is evident that fruit flies can, and do, use social information to inform many of their decisions.

One important context where social information seems to be used extensively by *D. melanogaster* females is in selecting suitable oviposition sites [5,8,22–25]. In the wild, there may be considerable heterogeneity in the availability and quality of oviposition sites that are available to female *D. melanogaster* [26,27], and the choice of oviposition environment can have profound consequences for their offspring's fitness [28,29]. *D. melanogaster* females may gain insight on the suitability of oviposition sites through direct observation of other females ovipositing [5,22], the presence of larvae [8] and/or the presence of chemical cues produced by conspecifics [25,30–32]. The reliance upon social cues by ovipositing *D. melanogaster* is not without potential pitfalls. First, by outsourcing one's information gathering to others, there is the risk of mimicking a conspecific's bad decisions. For instance, in a study by Golden and Dukas [8] when female flies were given a choice of laying their eggs on a site where nutrition was poor but contained social cues (the presence of larvae) or a site where the media contained 3 times more nutrition but lacked conspecific cues, the ovipositing females treated

them as being of equal quality, even though any offspring laid on the former patches did not have access to sufficient nutrients to survive. Similarly, Duménil et al. [32] observed that the presence of a mated female demonstrator on yeast-free media made this otherwise undesirable oviposition site equally-valued to later ovipositing females as yeasted media that has not been exposed to conspecifics. In *D. melanogaster*, the widespread use of social information may arise because there can be significant benefits of laying one's eggs at sites already chosen by conspecifics, as offspring growth and development may be enhanced through communal digestion via secreted enzymes [33,34], through increased local beneficial microbial activity [35–37] and/or via improved larval group foraging [38–40]. However, laying one's eggs near those of conspecifics can also pose risks arising from the competition for limited resources [8,41,42], increased exposure to toxic waste products [43], and the threat of cannibalism at many developmental stages [44–46]. Crucially, some of these dangers may potentially be mediated if females are able to oviposit at sites containing kin, as *D. melanogaster* larvae are more co-operative [39], and less cannibalistic [47,48] towards close relatives, compared to how they engage with unrelated conspecifics. Additionally, as there can be extensive intra-specific genetic variation *D. melanogaster* for nutritional preferences and dietary tolerances [49–51], preferentially laying eggs near those of close relatives may increase the likelihood of selecting the best micro-habitat for the success of one's own offspring. Thus, if females are able to differentiate social information produced by kin from those generated by other demonstrators, they might achieve greater fitness benefits (and avoid some of the risks) by selectively copying the choices made by relatives.

When considering the possibility that individuals might differentially use social cues produced by relatives versus non-relatives, one must also consider what (if any) cues could indicate kinship (or could be inferred to be reliable proxies of kinship). In *D. melanogaster* such information could conceivably be perceived from cuticular hydrocarbons (CHCs) and/or gut bacteria left behind by conspecifics. CHCs are long-chain hydrocarbons that are synthesized in insect oenocyte cells and become part of the cuticle's waxy coating, where they help prevent water loss, as well as also playing an important role in olfactory communication related to species recognition and mate choice decisions [25,52–56]. CHCs may provide accurate information about kinship as there exists a substantial genetic variation underlying these phenotypic traits within *D. melanogaster* populations [56–58]. Recent work has revealed that female flies transfer their own CHCs (along with male-specific compounds acquired during mating) to their laid eggs [25], which could conceivably be used as signals of potential kinship by subsequent ovipositing females.

In addition to CHC cues, olfaction-based decisions may be influenced by signals associated with differences in an individual's bacterial community [59,60]. While the *D. melanogaster* microbiome is relatively simple compared to other hosts [61], substantial variation exists between individuals/families from the same population in their microbial community composition [58,62]. Some of an individuals' gut flora is acquired from the microbial symbionts transferred by their mothers to the chorion of their eggs, which is subsequently consumed by the hatched offspring [63,64]. Thus, differences in cues produced by bacterial communities could conceivably (and independently of CHCs) be used to infer kinship [65,66]. An important, and potentially complicating factor, is that CHC phenotypes and/or microbiotic communities in *D. melanogaster* can also be influenced by the flies' developmental environment [30,58,59,65,67–69]. This raises the possibility that the phenomenon of "kin recognition" may be based on inherited cues of relatedness, on the recognition of similarity of environmental histories ("familiarity"), or some combination of these two factors, and that furthermore under some circumstances that these cues might provide contradictory or unreliable information about kinship. In a previous study that examined whether kinship mediated the

magnitude and consequences of male-male competition in adult flies, it was found that reductions in male-induced harm were *only* seen in groups of males that were both closely related to each other *and* 'familiar' by having developed in the same larval environment [70]. Lizé et al. [65] described how a male fruit fly's ability to strategically avoid mating with a sibling was impeded if they had both developed in the same environment. Similarly, Heys et al. [58] noted that the ability of fruit flies of both sexes to perceive the kinship status of their mate appeared to be dependent on the state of the mate's microbiome – with females only acting differently towards kin when those individuals had developed in the same media type (and had an intact microbiota), while a male's increased investment into mating with an unrelated female only occurred if she had been grown in a different environment (or whose microbiota had been disrupted with the use of antibiotics). The complicated nature of the interaction between developmental environment and relatedness was also explored by García-Roa et al. [68] who found that flies developing in different environments exhibited both different CHC and microbial profiles, that related flies had microbiotic communities of similar diversity (but not composition), and that diversity was correlated with CHC phenotypic profiles. Together, these studies highlight that socially-available kinship cues (or proxies thereof) may involve a number of independent and inter-related olfactory signals, whose expression may depend not only on relatedness but also on the specific nature of the developmental environment. As such, any study – such as ours – that explores the potential role of kin recognition in decision making, *must* explore how individuals respond to cues produced by demonstrators in which relatedness and familiarity are independently manipulated.

With these ideas and findings in mind, in this study we set out to examine whether female flies showed bias in their use of social information cues left behind by conspecifics when selecting ovipositing sites, and would differ in their use of egg-laying substrates that had previously been in contact with either kin or non-kin demonstrator females. Simultaneously, we also explored whether a female's response to potential kinship cues were conditional on the conspecifics' "familiarity", by concurrently manipulating the developmental environment from which the demonstrator females were obtained. Overall, the goal of this research is to better characterize copying behavior in *D. melanogaster* and to gain better insight into the factors that influence these oviposition site decisions.

## Materials and methods

### Population protocols and fly maintenance

All fruit flies used in our assays were obtained from one of three associated laboratory *D. melanogaster* populations: *Ives* (hereafter "IV"), "IV-*bw*", and "IV-*bwD*". The IV population was derived from a sample of mated females collected in 1975 from Amherst MA, USA sample population [71]. The IV-*bw* population was created by introgressing the brown-eye recessive allele, $bw^1$, into the genetic background of the IV population via 10 rounds of back-crossing, to create a competitor population that has > 99.9% of the IV genetic background [47]. The IV-*bwD* population was also created in the same fashion, by introducing the *bwD* allele (a dominant allele that also produces a brown-eyed phenotype) into an IV genetic background. The IV-*bw* and IV-*bwD* populations have been subject to periodic rounds of additional back-crossing to the IV population in an attempt to minimize their divergence from the base population, with the last session occurring approximately 50 generations prior to the start of this experiment. These three populations are each maintained at a census size of ~ 3500 adults per generation, incubated at 25°C incubation temperature with 60% humidity on a 12L:12D hour diurnal light cycle [72]. Flies are maintained in standard *Drosophila* vials (95mm H x 28.5mm OD) containing 10mL of banana/agar/killed-yeast food, dusted with live

yeast [71,73]. Populations are cultured on a 14-day cycle, wherein every fortnight adult flies are removed from their natal vials, mixed *en masse* under light $CO_2$ anaesthesia and distributed into vials containing new media for up to 18h, before being removed and the eggs laid are then trimmed by hand to a density of approximately 100 eggs per vial [72]. This study did not require approval from an ethics committee.

## Experiment 1: Test of the effects of conspecific relatedness and developmental environment on focal female behaviour and offspring production decisions

In our first experiment, we collected adult male and female flies as virgins (within 8h of eclosion from their pupae) from the IV population and housed them individually. We then created an initial 50 fly "families" by randomly combining single pairs of males and females into "mini-egg" chambers (Kartell 733/4 polyethylene 20 mL sample vials; 74.5mm H x 24.8mm OD; Fig S1a) whose lid contained juice/agar media [74] and was used as a "dish" for ovipositing. We left these chambers in the incubator for ~18h, before removing the adult flies and counting the number of eggs that had been laid overnight. We then evenly split each family's eggs among 2 different vials each containing 10ml of media, and then added a sufficient number of similarly-aged eggs obtained from the IV-*bw* population so that each vial ultimately contained ~100 eggs apiece. We returned these pairs of vials to the incubator where they were allowed to develop for the next 9 days, at which point we began collecting virgin females for use in the experiment. For each pair of vials, we haphazardly designated one vial as the "focal" vial, from which we collected 3 virgin females: a IV female ("the focal female"), a second IV female (the focal female's sister, raised in the same vial), and one brown-eyed IV-*bw* female (an unrelated "familiar" female, who had developed in the same vial as the focal female). From the other ("non-focal") vial in the pair, we collected 2 virgin females: an IV female (the focal female's sister, raised in a different "unfamiliar" developmental environment) and one brown-eyed IV-*bw* female (an unrelated female, raised in environment that was "unfamiliar" to the focal female). In this way we were able to obtain all possible combinations of demonstrator females that differed in their genetic relatedness and their potential environmental familiarity. We kept these females individually for up to 48h in test-tubes containing media to further verify their virgin status (by the absence of any fertilized eggs). Next, we introduced two similarly-aged adult IV males into the focal female's vial (so that all her offspring would express the wild-type red-eye phenotype). At the same time, we transferred all other females into their own mini-egg chambers whose dish lids contained standard banana media containing 3 IV-*bwD* males so that all offspring produced by demonstrators would express a *brown-eyed* eye phenotype and to ensure that any male-specific compounds transferred to the dish surface would be consistent across all dishes [25,31]. Females and males were left overnight to ensure sufficient time for mating to occur, and for ovipositing to begin. On the following morning all non-focal flies were removed from the mini-egg chambers, and sets of four dishes were affixed using adhesive putty to the bottom of a larger chamber, (hereafter "arena"), which consisted of an inverted plastic box (KIS Omni box, 20.3 x 15.9 x 9.6 cm) modified by the addition of mesh vent holes to box's outer edges (Fig S1b, [75]). A fifth dish (that had never been exposed to flies) was also placed in the arena before it was sealed shut, and the (lightly anesthetized) focal female was introduced via an access hole. Thus the five dishes inside each arena represent available oviposition sites for the focal female that potentially differ in their social information: The 'related/same' dish contains cues produced by a sister that developed in the same vial environment as the focal IV female; the 'unrelated/same' dish contains cues from an unrelated IV-*bw* female that developed in the same vial environment as

the focal IV female; the related/different' dish contains cues produced by a sister that developed in a different vial environment than the focal IV female; the 'unrelated/different' dish contains cues from an unrelated IV-*bw* female that developed in a different vial environment as the focal IV female; and finally the 'control/unexposed' dish contains no social information, as it had no prior exposure to flies (Fig S1c). In total we established 36 replicate arenas, which were housed in a quiet, well-lit room. Arenas were haphazardly rotated when placed on the counter to randomize any potential directional room effects, and were left undisturbed for 20 minutes following the introduction of the focal female. The arenas were then observed every 20 min for ~ 4.5h (starting at 10:30am), and for 4h on the next day (beginning at 9:30am) for a total of 26 observation sessions. During each observation session, the arenas were watched for a 5s time period, allowing for the identification of which of the 5 dishes (if any) the focal IV female was located. The identity of the different treatment dishes were unknown to the observers. Once all observations were complete, we unsealed the arena, discarded the focal female, and transferred the media in each dish into vials containing 10m of media. These labelled vials were sealed and returned to the incubator for 2 weeks. At that time, all eclosed adults were removed, and their eye colours were tallied. Red-eyed flies were the offspring of the focal female, while brown eye-flies ($wt + /bwD$ or $bw_1/bwD$) were produced by one of the conspecific females. We used the number of eclosed adult offspring as a proxy for the egg number laid, in this and subsequent experiments, as in the IV population at moderate larval densities, there is very high egg-to-adult survivorship [40].

## Experiment 2: Test of the effect of conspecific offspring abundance on focal female offspring production decisions

Based on the offspring production patterns observed in Experiment 1, we performed a series of complementary assays designed to test alternative hypotheses for the focal flies' potential oviposition biases. Thus, in this experiment, we set out to examine whether the number of offspring produced by conspecifics on a substrate was associated with the number offspring subsequently produced on the same dish by a focal female. We began by collecting mated IV-*bw* female flies from the lab's stock population and placed either 1, 2, or 0 individuals overnight into mini-egg chambers containing media in their dish lid. The next morning, one of each type of the dishes was affixed to the bottom of an arena (Fig S1d). Next, we added a single mated IV female to each of the 48 replicate arenas and left them undisturbed for 22h, before transferring the media in each dish to a vial containing 10ml of media. We returned those vials to the incubator and counted the number of wild-type and brown-eye offspring that had eclosed 12 days later.

## Experiment 3: Test of offspring production and behavioural biases of focal females towards cues from flies originating from the IV versus the IV-bw populations

In our third experiment we explored the alternate hypothesis that the offspring production patterns observed in Experiment 1 might reflect preferences of the IV-originating focal female for substrates that had been in contact with any other IV flies (regardless of their degree of immediate genetic consanguinity) and/or the avoidance of substrates that had come into contact with any IV-*bw* flies. For this assay we set up replicate vials containing 5 mated IV and 5 mated IV-*bw* females, that were allowed to oviposit for 18h before being discarded, and the number of eggs laid standardized to 100 apiece. These vials were placed in the incubator, and we collected a virgin red-eyed and a virgin brown-eyed female from each vial as they eclosed (thus both flies had experienced the same developmental environment). Each female was

then placed into their own mini egg chamber with 2 IV-*bwD* males overnight before being discarded, and the dish affixed to the inside of an arena, along with an unexposed/control dish (Fig S1e). An individual mated IV female obtained from the lab stock population was introduced to each of the arenas, and her location in the arena was periodically observed every 20 minutes over the course of 5h (for a total of 15 sessions). In total we established 72 replicate arenas, which were left undisturbed overnight, and the following morning media was removed from the dishes and transferred to vials containing 10ml of media. These vials were incubated for 12 days, and the number of offspring with different eye colours was recorded.

## Experiment 4: Revisiting the effect of conspecific relatedness on offspring production and behavioural decisions

Based on the outcomes of our second and third experiments, we set about to directly compare the behaviour and offspring production of focal females towards substrates that had been exposed to their sibling or to an unrelated IV-derived female. As in experiment 1 we collected the focal female's parents from the IV population, mated them, placed them into mini-egg chambers overnight, then counted and split their eggs amongst 2 vials, with additional similarly-aged IV-*bw* eggs added to bring the density to a total of 100 eggs apiece. From each of the pairs of vials we collected virgin females as they eclosed and kept them individually. One of the vials was arbitrarily designated as that which would yield the focal female (who was collected as a virgin, then housed with 2 IV males from the stock population overnight), and virgin IV female from the other vial (*i.e.,* the focal female's sister raised in a different environment) was housed with 2 IV-*bwD* males overnight in a mini egg chamber. We also haphazardly chose one of the virgin IV females collected from one of the other sets of vials (thus an unrelated female, raised in a different environment) and housed her with 2 IV-*bwD* males overnight in a mini egg chamber. These dishes (along with a control/unexposed dish) were placed into an arena (Fig S1f), into which we added the mated focal females. In each of the 84 replicate arenas, the location of the focal female was periodically observed every 20 minutes over the course of 5h (for a total of 15 sessions). Next, the chamber was left undisturbed overnight and the following morning, we transferred the media from each dish into a new vial containing 10ml of media which were incubated for an additional 12 days. The eye colour of all adult flies present in these vials were tallied and recorded.

## Statistical Analyses

All data analyses were conducted in the R statistical computing environment (version 4.0.3, [76]). In all of our experiments we compared the frequencies of visits by focal females to the five different lids in each arena. As these data were zero-inflated (determined using diagnostic functions in the *DHARMa* package [77] we constructed zero-inflated binomial mixed models using the *glmmTMB* function in the *glmmTMB* package [78] where the independent fixed factor was the dish treatment category, and the arena that contained them was included as a random effect. We determined whether the treatment means were significantly different from each other with likelihood-ratio chi-square tests using the *Anova* function in the *car* package [79], which was followed, if necessary, by post-hoc tests with Tukey method p-value adjustments executed using the *emmeans* function in the *emmeans* package [80] to determine the specific location of differences between treatments. For experiment 1 we also used a two-way, zero-inflated *glmmTMB* model to determine if there was any difference in patch visitation that varied with relatedness and/or familiarity treatments, by using the same response data (excluding observations made on the control dish, as this treatment had – by definition – received no prior conspecific contact). The demonstrator female's degree of kinship, their

environmental origin, and the interaction between these two factors were the independent fixed factors in this GLMM model, and arena was the random effect. The significance of these effects was determined using a likelihood-ratio chi-square test using the *Anova* function.

For all our experiments we (separately) analyzed the total number of red-eyed and brown-eyed adults that successfully eclosed from the different lids in each arena using zero-inflated *glmmTMB* GLMMs, with poisson error distributions, where the independent fixed factor was the dish treatment category, and arena was the random effect. For each model we determined whether the treatment means were significantly different from each other using likelihood-ratio chi-square tests implemented using the *Anova* function, and post-hoc tests were executed, if necessary, using the *emmeans* function, with Tukey method p-value adjustments, to determine the specific location of differences between treatments. As above for experiment 1 data, we also set out to determine if there was any difference in offspring number associated with relatedness and/or familiarity treatments, by excluding data collected from the control dish treatment, and analyzing the rest of the data set using a 2-way zero-inflated *glmmTMB* GLMM with poisson errors. The demonstrator female's degree of kinship, their environmental origin, and the interaction between these two factors were the independent fixed factors in this model, and arena was included as a random effect. The significance of these effects was determined using a likelihood-ratio chi-square test using the *Anova* function, and, if necessary, post-hoc tests were conducted using the *emmeans* function, with Tukey method p-value adjustments, to determine the specific location of differences between treatments.

In experiment 2 we also compared the frequencies of dishes that yielded zero focal offspring using the *chisq.test* function to conduct a Pearson's Chi-squared test, then examined the specific differences between cells in the contingency table using the *chisq.posthoc.test* function in the *chisq.posthoc.test* package [81].

## Results

### Experiment 1: Test of the effects of conspecific relatedness and developmental environment on focal female behaviour and offspring production decisions

In our first experiment, we compared the cumulative number of visits that *D. melanogaster* focal females made to five different dishes, as well as the number of offspring that eclosed as adults from each of those dishes. Our analysis found that those females exhibited heterogeneity in their dish associations (LR$\chi^2$ = 20.12, df = 4, p = 0.0005), and while they visited the control/unexposed dishes less frequently than the demonstrator dishes, within these four treatment groups, they did not express any significant bias (Fig S2). Offspring production patterns indicated that the females' oviposition patterns were also heterogeneous (Fig 1a; LR$\chi^2$ = 123.92, df = 4, p = 7.77x10$^{-26}$), with significantly more red-eyed focal offspring eclosing from those dishes that had previously been in contact with a related demonstrator individual than from either the control/unexposed dish or either of the dishes whose prior occupant was an unrelated individual, and the greatest number of offspring eclosing from those dishes that had been exposed to a related conspecific, raised in a different vial. Our focused two-way GLMM confirmed that the amount of oviposition activity on a dish was significantly associated with the demonstrator's kinship (LR$\chi^2$ = 84.01, df = 1, p = 4.91x10$^{-20}$) with more offspring eclosing from 'related' dishes ($\bar{x}$ = 4.97; 95% CI: 3.57, 6.92) than 'unrelated' dishes ($\bar{x}$ = 0.19; 95% CI: 0.08, 0.42), there was a small but significant effect of developmental environment (LR$\chi$2 = 9.38, df = 1, p = 0.0021), with slightly more offspring eclosing from 'different' dishes ($\bar{x}$ = 1.08; 95% CI: 0.57, 1.84) than 'same' dishes ($\bar{x}$ = 0.93; 95% CI: 0.52, 1.67), while the interaction between kinship and developmental environment was not significant (LR$\chi^2$

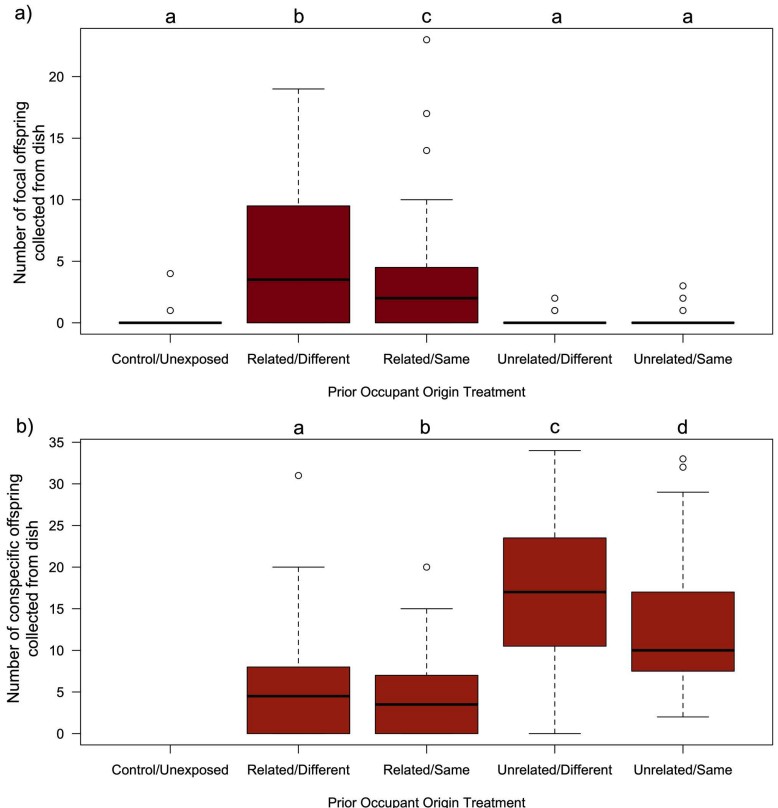

**Fig 1. More focal female offspring produced on dishes where the prior occupant was a related female, while more conspecific offspring were produced on dishes where the female was unrelated.** Boxplots illustrating data collected in from the 36 replicate arenas in the first experiment of a) the total number of wild-type (*red-eyed*) adult flies that eclosed from eggs laid by the focal *D. melanogaster* females on 5 different media dishes, b) the total number of *brown-eyed* adult flies that eclosed from those same 5 media dishes, which had been laid by the prior conspecific occupants on the dish. The boxes enclose the middle 50% of data (the inter-quartile range, IQR), with the thick horizontal line representing the location of median. Data points> ± 1.5*IQR are designated as outliers. Whiskers extend to largest/smallest values that are not outliers, indicated as closed circles. The results of Tukey HSD post-hoc tests comparing group mean are indicated by letters, where groups that do not share the same letter are considered statistically different at the α=0.05 level. The control/unexposed dishes by definition had not had any prior exposure to a female capable of producing *brown-eyed* offspring, and thus those dishes never yielded any offspring with that phenotype.

= 1.20, df = 1, p = 0.27). We also noted that the number of demonstrator offspring were not homogeneous across the four treatment groups (Fig 1b), with more brown-eyed offspring eclosing from dishes whose prior occupant was a unrelated female (IV-*bw*, mated to a IV-*bwD* male; x̄ = 14.49; 95% CI: 13.03, 16.10) than those that had previously been exposed to a related female (IV, mated to a IV-*bwD* male; x̄ = 6.83; 95% CI: 5.95, 7.83; LRχ² = 147.93, df = 1, p = 4.90x10⁻³⁴), and fewer offspring eclosing from vials where the demonstrator originated from the same vial (x̄ = 8.55; 95% CI: 7.53, 9.71) than if they were collected from a different vial (x̄ = 11.57; 95% CI: 10.28, 13.02; LRχ² = 30.83, df = 1, p = 2.82x10⁻⁸), yet there was no significant interaction between these two factors (LRχ² = 0.001, df = 1, p = 0.975).

## Experiment 2: Test of the effect of conspecific offspring abundance production on focal female offspring production decisions

In our second experiment, we set out to test the alternate hypothesis that the biased patterns of *red-eyed* offspring produced by our focal females was influenced by the abundance of

*brown-eyed* conspecific offspring already present on the dishes. In this assay either zero, one or two demonstrator females were exposed to a dish prior to being offered as a potential oviposition site to a mated focal female. More demonstrators on a dish yielded more *brown-eyed* offspring: zero *brown-eyed* offspring eclosed from the control dishes, and two demonstrator females produced, on average more *brown-eyed* offspring ($\bar{x} = 13.53$; 95% CI: 11.2, 16.3) than a single demonstrator female ($\bar{x} = 8.68$; 95% CI: 7.0, 10.8; $LR\chi^2 = 36.14$, df = 1, p = $1.84 \times 10^{-9}$; Fig 2a). Despite the dramatic differences in the number of demonstrator offspring that eclosed between the three dishes, there was no significant difference in the number of *red-eyed* focal offspring that eclosed from the three dishes ($LR\chi^2 = 1.99$, df = 2, p = 0.371; Fig 2b). Furthermore, dishes that had been exposed to two demonstrator females were significantly *less* likely to yield *zero* focal female offspring (8/48) than those dishes that had been exposed to either one demonstrator female (25/48) or from control dishes (32/48; $\chi^2 = 13.43$, df = 2, p = 0.0012).

## Experiment 3: Test of offspring production and behavioural biases of focal females towards cues from flies originating from the IV versus the IV-bw populations

In our third experiment, we set out to test the alternate hypothesis that the patterns of *red-eyed* offspring produced by our focal females was influenced by the population of origin of the demonstrator females, and not their degree of kinship. This involved comparing the cumulative number of visits that *D. melanogaster* focal females made to three different dishes (a control/unexposed dish, a dish where the demonstrator was an unrelated IV female, or a dish whose demonstrator was an unrelated IV-*bw* female), as well as the number of offspring that eclosed as

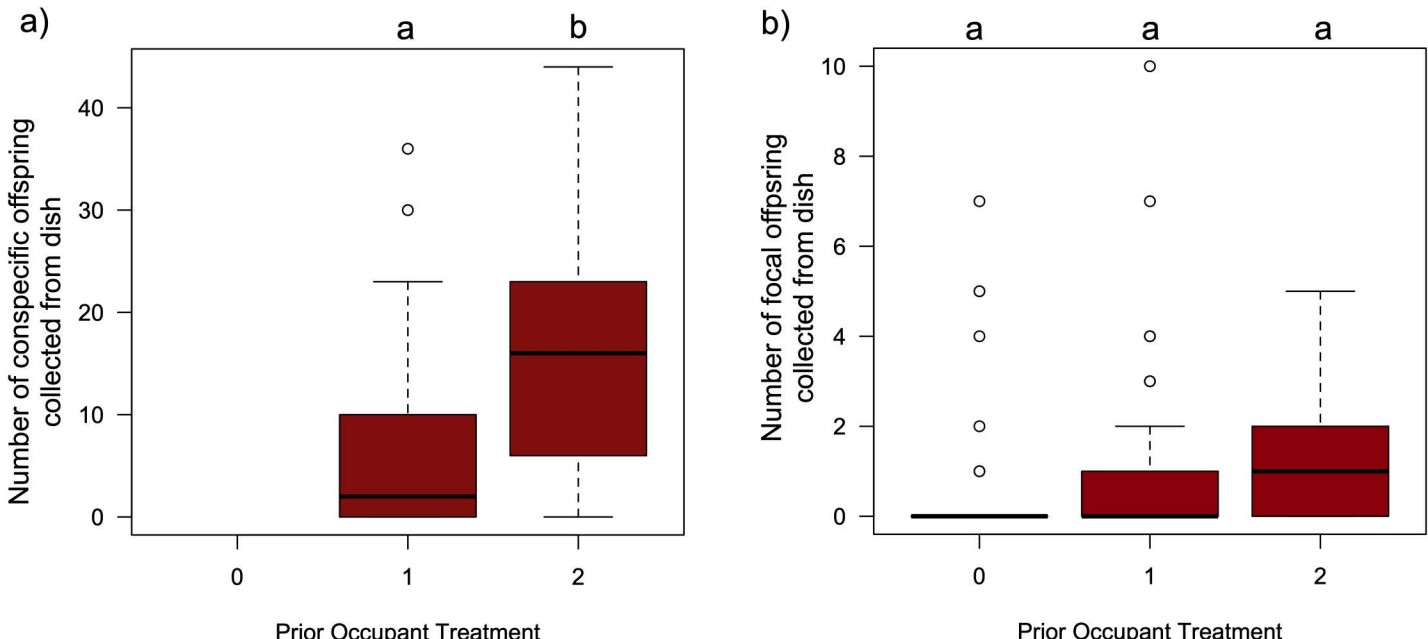

**Fig 2. No evidence that focal female offspring production on dishes is inhibited by increased abundance of conspecific offspring.** Boxplots illustrating the data collected from the 48 replicate arenas in the second experiment of a) the number of *brown-eyed* adult flies that eclosed from dishes that had been exposed to 0, 1 or 2 conspecific females, b) the number of *wild-type* (red-eyed) adult flies that eclosed from the same dishes that had been laid by the focal female *D. melanogaster*. Boxplot components as in Figure 1. The results of Tukey HSD post-hoc tests comparing group mean are indicated by letters, where groups that do not share the same letter are considered statistically different at the α=0.05 level. The control/unexposed dishes by definition had not had any prior exposure to a female capable of producing *brown-eyed* offspring, and thus those dishes never yielded any offspring with that phenotype.

adults from each of those dishes. Our analysis found that focal females exhibited heterogeneity in their dish associations (LRχ$^2$ = 144.8, df = 2, p = 3.61 x10$^{-32}$) with the control/unexposed dishes receiving significantly fewer visits (x̄ = 0.05; 95% CI: 0.03, 0.07), than either the dishes that had been exposed to the unrelated IV demonstrator (x̄ = 0.31; 95% CI: 0.25, 0.36) or the dishes that had been exposed to the unrelated IV-*bw* demonstrator (x̄ = 0.25; 95% CI: 0.21, 0.31), with no significant difference in visitation rates to the two demonstrator dishes (t = 21.96, df = 1, p = 0.07; Fig S3). The distribution of focal female offspring was also heterogenous across the three dish types (Fig 3b; LRχ$^2$ = 139.27, df = 2, p = 5.72 x10$^{-31}$). More focal female offspring eclosed from the unrelated IV dishes (x̄ = 8.84; 95% CI: 7.45, 10.49) than they from either the control/unexposed dishes (x̄ = 2.32; 95% CI: 1.49, 3.61), or the unrelated *IV-bw* dishes (x̄ = 3.09; 95% CI: 2.44, 3.91). The mean number of *red-eyed* offspring that were collected from the *IV-bw* dish was not significantly different from the number collected from the control/unexposed dish (t = 1.31, df = 1, p = 0.39). We also noted that the number of eclosed demonstrator offspring were significantly different between dish types (Fig 3a; LRχ$^2$ = 89.42, df = 1, p = 3.82 x10$^{-20}$), with more brown-eyed offspring eclosing from vials whose prior occupant was an unrelated IV female (mated to a IV-*bwD* male; x̄ = 12.33; 95% CI: 10.83, 14.05) than those that that had previously been exposed to an unrelated IV-*bw* female (also mated to a IV-*bwD* male; x̄ = 6.53; 95% CI: 5.55, 7.69).

## Experiment 4: Revisiting the effect of conspecific relatedness on offspring production and behavioural decisions

In our fourth experiment, we set out (again) to compare the patterns of dish visitation and offspring production by focal females on three different dishes (a control/unexposed dish, a dish

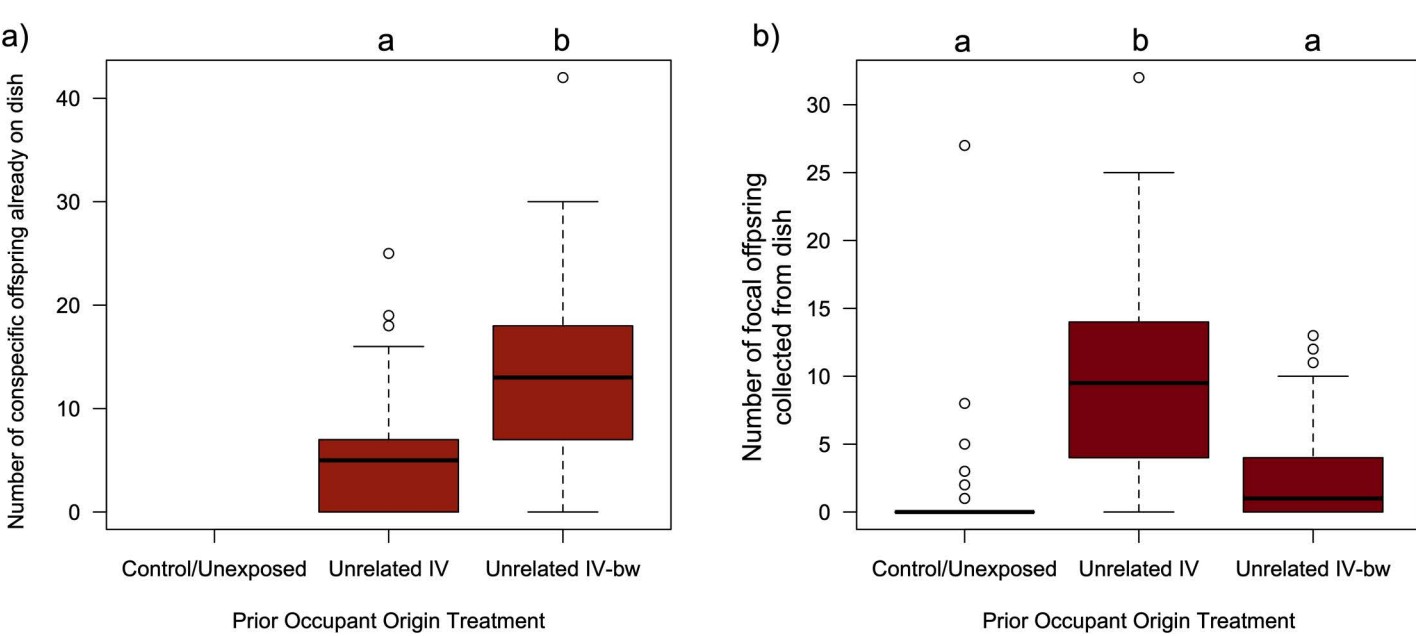

**Fig 3. Focal female offspring production is higher on dishes where the unrelated demonstrator originates from the same (IV) population than a different (IV-bw) population.** Boxplots illustrating data collected from the 72 replicate arenas in third experiment of a) the number of *brown-eyed* adult flies that eclosed from dishes that had been exposed to either an unrelated IV female (mated to a IV-*bwD* male), a unrelated IV-*bw* female (mated to a IV-*bwD* male), or neither b) the number of *wild-type* (red-eyed) adult flies that eclosed from the same dishes that had been laid by the focal female *D. melanogaster.* Boxplot components as in Figure 1. The control/unexposed dishes by definition had not had any prior exposure to a female capable of producing *brown-eyed* offspring, and thus those dishes never yielded any offspring with that phenotype. The results of Tukey HSD post-hoc tests comparing group mean are indicated by letters, where groups that do not share the same letter are considered statistically different at the α=0.05 level.

whose demonstrator was a related IV female, or a dish where the demonstrator was an unrelated IV female). The focal females exhibited bias on their dish visitation rates (Fig S4; LRχ2 = 12.81, df = 2, p = 0.0017), as they were significantly fewer visitations to the control/unexposed dishes (x̄ = 0.05; 95% CI: 0.03, 0.07), than to related IV demonstrator dishes (x̄ = 0.08; 95% CI: 0.06, 0.12) or to the unrelated IV dishes (x̄ = 0.11; 95% CI: 0.08, 0.14). There was no significant difference in visitation rates between the two demonstrator groups (t = 1.33, df = 1, p = 0.38). The distribution of focal female offspring was heterogenous across the three dish types (Fig 4; GLM: LRχ² = 368.28, df = 2, p = 1.07x10⁻⁸⁰). Significantly more focal female offspring eclosed from the related IV dishes (x̄ = 11.46; 95% CI: 9.96, 13.19) than from the unrelated IV dishes (x̄ = 7.47; 95% CI: 6.43, 8.68), and both demonstrator dishes yielded more focal female offspring than did the control/unexposed dishes (x̄ = 0.48; 95% CI: 0.33, 0.69). There was a small, but statistically significant difference in the number of demonstrator offspring eclosing from the related IV dishes (x̄ = 11.3; 95% CI: 10.32, 12.4) and from the unrelated IV dishes (x̄ = 10.1; 95% CI: 9.14, 11.1), LRχ² = 6.59, df = 1, p = 0.01; Fig S5).

## Discussion

The decision where to oviposit can be very important, especially in those species where there is no post-laying parental care [28,29]. Fruit flies, *Drosophila melanogaster*, like many other species, use social information in their decision-making [11,12]. While there are benefits of laying one's eggs in proximity to those of others, there are also risks – both of which might be mediated by the degree of kinship between conspecifics. In this study, we set out to examine the possibility that female flies might selectively use cues produced by close relatives, presumably to increase their offspring's chance of success. In doing so, we simultaneously explored the factors that might influence how flies might infer kinship, as well as investigate alternate

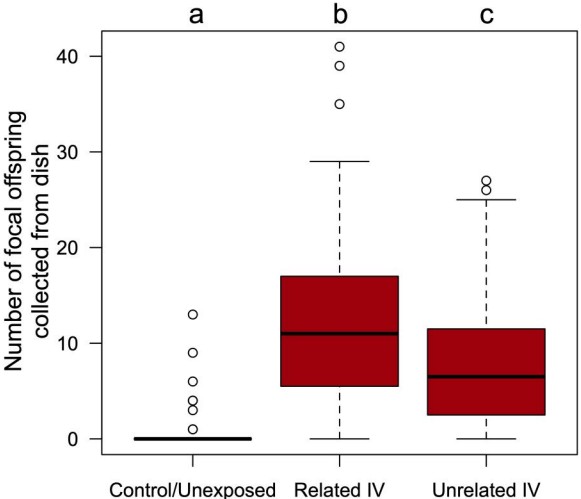

**Fig 4. More focal female offspring produced on dishes where the prior (IV-originating) occupant was a related female.** Boxplot illustrating data collected from the 84 replicate arenas in the fourth experiment of the number of *wild-type* (red-eyed) adult flies that had been laid by the focal female *D. melanogaster* that eclosed from dishes that had been exposed to either a related IV female (mated to a IV-*bwD* male), an unrelated IV female (mated to a IV-*bwD* male), or neither. Boxplot components as in Figure 1. The results of a Tukey HSD post-hoc test comparing group mean is indicated by letters, where groups that do not share the same letter are considered statistically different at the α=0.05 level.

hypotheses about social information usage. Our assays confirm the extensive use of social cues by *D. melanogaster* females, and furthermore collectively suggest that they are capable of both kinship discrimination and distinguishing between conspecifics from different populations. These results add to the growing body of knowledge about the sophisticated social information use by this model species [12].

In all assays focal females associated more with those substrates that had come into contact with conspecifics than with the unexposed/control dishes (Figs S2, S3 and S4), clearly illustrating the importance of social cues in their decision-making, and consistent with numerous previous studies [*e.g.,* 22,24,32]. However, not all conspecific cues elicited the same responses. In the first assay, we compared the offspring production patterns of focal females on dishes exposed to a range of different demonstrators. By splitting sets of full sibling eggs between two different vials to develop, and collecting both adult relatives and non-relatives from each of the pairs of vials, we set out to test if and how *D. melanogaster* females might distinguish kin cues. We observed that dishes that had been in previous contact with a related demonstrator female yielded considerably more focal female offspring that those where the demonstrator was non-kin. Interestingly, dishes where the demonstrator came from the same vial as the focal vial produced fewer focal female offspring than those when the demonstrator had developed in the same vial, however the size of this difference was much smaller, and there was no significant interaction between demonstrator kinship and origin factors (Fig 1a). These results suggest that ovipositing females are selective in their use of social information, and favor cues produced by their relatives, while perceived environmental familiarity does not enhance oviposition site use. Determining the specific cues being used by focal females is beyond the scope of this study as there are numerous ways that social information may be obtained. When ovipositing, female fruit flies transfer gut bacteria, CHCs, and male-specific compounds acquired during copulation to the surface of their eggs and the media [25,63,64]. The expression of these olfactory cues can be influenced by both an individual's genotype and/or their developmental environment [30,58,59,65,67–69]. The pattern of significant differences associated with the developmental environment (Fig 1a) might suggest that "familiarity" – and the microbiotic cues that might indicate kinship – is either unimportant or misleading. However, it is important to note that both experiment vial environments were similar (same media and larval density, which may have reduced the potential for differences in cues to develop. A previous study found that *D. melanogaster* females were more likely to oviposit near eggs laid by other females whose developmental diet matched their own, regardless of their population of origin or the composition of the diet itself [24]. Thus, one should not necessarily conclude that our results indicate that inherited cues are the only way in which egg kinship was/can be recognized by the focal females in this species, and future studies exploring this idea across a range of divergent environments are likely to expand our understanding of how social information is produced and interpreted in *D. melanogaster*.

While the observed pattern of focal female offspring production is consistent with kin-associated biases, other processes could also have resulted in the differential outcomes, and so to validate or rule out these alternate hypotheses we conducted several complementary assays. We had been initially struck with the distinctive pattern of offspring production by the demonstrator females (Fig 1b), wherein there were more *brown-eyed* conspecific offspring collected from dishes where the demonstrator was unrelated than when the demonstrator was related. Perhaps our focal females weren't deciding to lay their eggs on dishes that contained their kin's offspring, but instead were avoiding dishes where there were already lots of eggs (and a heightened risk of competition and/or cannibalism)? We explored this possibility in our second assay, where focal females could choose between dishes with dramatically different number of demonstrator offspring (Fig 2a). If females were avoiding dishes where there were

too many conspecific eggs already present, we would expect to see more focal offspring eclosing on the "one female" treatment than on the "two females" treatment. In fact, we observed no significant difference in focal female offspring production between the two groups (and if anything, there were fewer cases where we observed *zero* focal female offspring produced when there were more demonstrators; Fig 2b). Previous work by Golden and Dukas [8] found that female flies given a choice between a control and a demonstrator dish (which had been exposed to 5 or 20 larvae), did not appear to discriminate against the demonstrator dishes that possessed cues indicating greater competition during offspring development. Thus, we are fairly confident that the differences in focal female offspring production patterns observed in experiment 1 were not due to differences in the quantity of demonstrator cues present on a dish. As a side note, we suggest that the differences in the number of demonstrator offspring observed in that assay is a side effect of the way in which our demonstrator females were collected (*see* Fig S6 *caption for details*).

The second alternate hypothesis we explored was that the biased pattern of oviposition made by the focal females in the first experiment on the two "related" dishes, was not due to the perceived kinship of the demonstrators, but instead was because those demonstrators originated from the same population (IV), and the demonstrators for the two "unrelated" dishes were from a different population (IV-*bw*). While IV and IV-*bw* are maintained following the same protocols, and IV-*bw* is periodically back-crossed to IV, perceptible differences in the cues present in these isolated populations (or their microbiotic communities) may have arisen via drift and/or selection. Thus, in our third experiment we compared the behaviour and offspring production of focal females exposed to (unrelated) IV or IV-*bw* demonstrator females. We found that focal females produced more offspring on the former dishes than they did on the latter (and virtually none on the control/unexposed dishes). This suggests that at least some of the patterns of offspring production seen in our first experiment may not necessarily be the result of recognition of kinship *per se*, but instead with other cue(s) that differ between the IV and IV-*bw* populations. Thus, based on the insights gained in our third assay, we set out to re-examine the social information use of *D. melanogaster* females in our fourth assay where females were given a choice of oviposition sites that either had no prior exposure (control), a demonstrator sister (raised in a different vial), or an unrelated demonstrator female (also from the IV population and raised in a different vial). We observed that focal females still biased their offspring towards the dishes that had previously been in contact with their sisters, albeit to a lesser extent than seen in experiment 1. This provides additional support to the hypothesis that female *D. melanogaster* are both capable of distinguishing between cues produced by kin and non-kin, and that these cues are used in their social decision-making. In doing so, they may be trying to obtain the potential benefits of developing in a group [33–38,73], while reducing the potential risks from conspecifics [38,47,48].

Collectively, our results illustrate the subtle and plastic nature of social information use by female *D. melanogaster*: Given a choice, a mated female *D. melanogaster* will be more likely to produce offspring on a dish that has previously been in contact with any conspecific, compared to an socially unexposed surface, there will be more offspring production associated with substrates that have been in previous contact with an unrelated mated female from the same population than one exposed to a mated female from a different population, and that focal if a substrate has been exposed to a mated sister, that site will yield more focal offspring than if the demonstrator is an unrelated individual. At the same time, mated female *D. melanogaster* do not appear to show any bias in their egg production against dishes that differ in the number of conspecific eggs (and their associated biochemical cues) present. Together, these pattens illuminate that *D. melanogaster* females are sensitive to differences in the cues produced by conspecifics and add to the growing appreciation of social decision-making in this model species.

## Supporting information

**S1 Figure. Drawings illustrating a) the "mini-egg" chambers used to collect eggs and/or create media that exhibited social cues.** b) an observation arena into which the focal ovipositing female was introduced and monitored. At the bottom of the chamber the lids that exhibited different kinds of social information were affixed. **c)** the arrangement of dishes in the first experiment. In this experiment there were four lids that exhibited social information and the 'control/unexposed' dish that contained no social information (as it had no prior exposure to flies). **d)** the arrangement of the two dishes with social information and the control dish in the second experiment. **e)** the arrangement of the two dishes with social information and the control dish in the third experiment. **f)** the arrangement of the two dishes with social information and the control dish in the fourth experiment.
(PDF)

**S2 Figure. Focal females visited demonstrator-exposed dishes more frequently than to control/unexposed dishes.** Boxplots illustrating the cumulative number of observations made across 26 sessions in the first experiment in which a focal female *D.melanogaster* was observed on the surface of one of 5 different media dishes present in the 36 replicate arenas. The boxes enclose the middle 50% of data (the inter-quartile range, IQR), with the thick horizontal line representing the location of median. Data points$> \pm 1.5 * IQR$ are designated as outliers. Whiskers extend to largest/smallest values that are not outliers, indicated as closed circles. The results of a Tukey HSD post-hoc test comparing group mean is indicated by letters, where groups that do not share the same letter are considered statistically different at the α=0.05 level.
(PDF)

**S3 Figure. Focal females visited demonstrator-exposed dishes more frequently than to control/unexposed dishes.** Boxplots illustrating the cumulative number of observations made across 26 sessions in the third experiment in which a focal female *D. melanogaster* was observed on the surface of one of 3 different media dishes present in the 72 replicate arenas that had previously been exposed to either an unrelated IV female (mated to a IV-*bwD* male), an unrelated IV-*bw* female (mated to a IV-*bwD* male), or neither. Boxplot components as in Figure S2. The results of a Tukey HSD post-hoc test comparing group mean is indicated by letters, where groups that do not share the same letter are considered statistically different at the α=0.05 level.
(PDF)

**S4 Figure. Focal females visited demonstrator-exposed dishes more frequently than to control/unexposed dishes.** Boxplots illustrating the cumulative number of observations made across 26 sessions in the fourth experiment in which a focal female *D. melanogaster* was observed on the surface of one of 3 different media dishes present in the 84 replicate arenas that had previously been exposed to either a related IV female (mated to a IV*bwD* male), an unrelated IV female (mated to a IV-*bwD* male), or neither. Boxplot components as in Figure S2. The results of a Tukey HSD post-hoc test comparing group mean is indicated by letters, where groups that do not share the same letter are considered statistically different at the α=0.05 level.
(PDF)

**S5 Figure. Number of conspecific offspring collected from demonstrator-exposed dishes.** Boxplots illustrating the number of *brown-eyed* adult flies that eclosed from dishes that had been exposed to either a related IV female (mated to a IV-*bwD* male), an unrelated IV female (mated to a IV-*bwD* male), or neither in the fourth experiment. There was a small (Cliff's

Delta effect size = 0.15) but statistically significant difference in the mean number of offspring collected from these dishes. Boxplot components as in Figure S2. The control/unexposed dishes my definition had not had any prior exposure to a female capable of producing *brown-eyed* offspring, and thus those dishes never yielded any offspring with that phenotype. (PDF)

**S6 Figure. Schematic diagram depicting hypothetical distribution of female *D. melanogaster* eclosion times in a mixture of eggs collected at the same time from the IV and the IV-*bw* populations (where we assume are randomly sampled, and that both populations have identical distributions, means and variances).** While both subset means are the same, as there is a greater number of IV-*bw* eggs in the vial than IV eggs, the larger sample size of the former will likely contain individuals exhibiting a wider range of phenotypes than the latter. The unanticipated consequence of this being that when sampling for virgin female flies, there is a greater probability of collecting a suitable IV-*bw* before encountering a suitable IV female. As body size is negatively correlated with eclosion time (Partridge et al. 1987), and positively correlated with female fecundity (Lefranc and Bundgaard 2000, Long et al. 2009), it is conceivable that the IV-*bw* and the IV females collected and used as demonstrators differed in their capacity to produce offspring, resulting in the conspecific patterns seen in Figure 1b and 3b. It is worth noting that in our fourth experiment, where only *red-eyed* females were collected from vials containing both IV and IV-*bw* eggs that there was only a small difference in the number of demonstrator offspring that were collected (Figure S5), which is consistent with the hypothesis described above. (PDF)

## Author contributions

**Conceptualization:** Emily Rakosy, Tristan A.F. Long.

**Data curation:** Emily Rakosy, Tristan A.F. Long.

**Formal analysis:** Tristan A.F. Long.

**Funding acquisition:** Tristan A.F. Long.

**Investigation:** Emily Rakosy, Sanduni Talagala, Tristan A.F. Long.

**Methodology:** Emily Rakosy, Tristan A.F. Long.

**Project administration:** Emily Rakosy.

**Resources:** Tristan A.F. Long.

**Supervision:** Tristan A.F. Long.

**Writing – original draft:** Emily Rakosy, Tristan A.F. Long.

**Writing – review & editing:** Emily Rakosy, Sanduni Talagala, Tristan A.F. Long.

## Acknowledgements

We would like to thank Simran Mann, Lukas Ghiglione, and Harleen Taneja of the #DrosLife Lab for their fly-pushing, behavioral observations and camaraderie. Natasha B. Gallo and three anonymous reviewers provided helpful feed-back and constructive comments. Ruby Lindsay and Michael Steeleworthy of the Wilfrid Laurier University Library are thanked for their help with data archiving. This work was conducted at Wilfrid Laurier University, which exists on the traditional territory of the Neutral, Anishnawbe, and Haudenosaunee peoples.

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
