## [Decision Letter · Decision Letter 0]

2 Oct 2024

PONE-D-24-34027On the use of kinship and familiarity associated social information in mediating Drosophila melanogaster oviposition decisionsPLOS ONE

Dear Dr. Rakosy,

Thank you for submitting your manuscript to PLOS ONE. After careful consideration, we feel that it has merit but does not fully meet PLOS ONE’s publication criteria as it currently stands. Therefore, we invite you to submit a revised version of the manuscript that addresses the points raised during the review process.

**The reviewer's reports are included at the bottom of this email or can be accessed, together with a copy of this decision letter, by logging on to the journal editorial manager. **

**As you will note, the reviewers raise a number of substantial criticisms that prevent me from accepting the manuscript at this stage. Both reviewer's solicit additional clarification on certain outstanding issues including experimental rationale, sample number, data analysis, etc., and provide constructive feedback to improve the clarity and presentation of the article.  Additionally, we note that the online repository link for deposited data is not working. **

We look forward to receiving your revised manuscript.

Kind regards,

Madhumala Sadanandappa, PhD

Academic Editor

PLOS ONE

Journal Requirements: When submitting your revision, we need you to address these additional requirements. 1. Please ensure that your manuscript meets PLOS ONE's style requirements, including those for file naming. The PLOS ONE style templates can be found at https://journals.plos.org/plosone/s/file?id=wjVg/PLOSOne_formatting_sample_main_body.pdf and https://journals.plos.org/plosone/s/file?id=ba62/PLOSOne_formatting_sample_title_authors_affiliations.pdf 2. Please update your submission to use the PLOS LaTeX template. The template and more information on our requirements for LaTeX submissions can be found at http://journals.plos.org/plosone/s/latex. 3. Thank you for stating the following in the Acknowledgments Section of your manuscript: "We would like to thank Simran Mann, Lukas Ghiglione, and Harleen Taneja of the #DrosLife Lab for their fly-pushing, behavioral observations and camaraderie. Natasha B. Gallo provided helpful feed-back and constructive comments. Ruby Lindsay and Michael Steeleworthy of the Wilfrid Laurier University Library are thanked for their help with data archiving. T.A.F.L. was funded with a Natural Sciences and Engineering Research Council Discovery grants (RGPIN740 2016-06133 and RGPIN-2022-03988). This work was conducted at Wilfrid Laurier University, which exists on the traditional territory of the Neutral, Anishnawbe, and Haudenosaunee peoples." We note that you have provided funding information that is not currently declared in your Funding Statement. However, funding information should not appear in the Acknowledgments section or other areas of your manuscript. We will only publish funding information present in the Funding Statement section of the online submission form. Please remove any funding-related text from the manuscript and let us know how you would like to update your Funding Statement. Currently, your Funding Statement reads as follows: "TAFL received support in the for of Natural Sciences and Engineering Research Council (https://www.nserc-crsng.gc.ca) Discovery grants (RGPIN-2016-06133 and RGPIN-2022-03988). The funding agency played no role in the study design, data collection and analysis, decision to publish, or preparation of the manuscript" Please include your amended statements within your cover letter; we will change the online submission form on your behalf. 4. Please note that in order to use the direct billing option the corresponding author must be affiliated with the chosen institute. Please either amend your manuscript to change the affiliation or corresponding author, or email us at plosone@plos.org with a request to remove this option. 5. We notice that your supplementary figures are included in the manuscript file. Please remove them and upload them with the file type 'Supporting Information'. Please ensure that each Supporting Information file has a legend listed in the manuscript after the references list. 6. Please review your reference list to ensure that it is complete and correct. If you have cited papers that have been retracted, please include the rationale for doing so in the manuscript text, or remove these references and replace them with relevant current references. Any changes to the reference list should be mentioned in the rebuttal letter that accompanies your revised manuscript. If you need to cite a retracted article, indicate the article’s retracted status in the References list and also include a citation and full reference for the retraction notice.

Reviewers' comments:

Reviewer's Responses to Questions

**Comments to the Author**

1. Is the manuscript technically sound, and do the data support the conclusions?

Reviewer #1: Yes

Reviewer #2: Yes

2. Has the statistical analysis been performed appropriately and rigorously? 

Reviewer #1: Yes

Reviewer #2: Yes

3. Have the authors made all data underlying the findings in their manuscript fully available?

Reviewer #1: No

Reviewer #2: Yes

4. Is the manuscript presented in an intelligible fashion and written in standard English?

Reviewer #1: Yes

Reviewer #2: Yes

5. Review Comments to the Author

Reviewer #1: I really enjoyed reading this paper – I thought it was clearly written and the results support the conclusions. The authors address the interesting question of whether relatedness of conspecific females plays a role in how Drosophila use public information to inform egg laying decisions. This is a novel study, which finds some evidence that relatedness and population of origin of conspecifics are likely important factors in female patch preferences. The authors also try to separate out some confounding factors of their main experiment. These latter assays do somewhat strengthen the hypothesis that relatedness is important, but further follow up experiments are probably required to strongly conclude that females are responding to relatedness of conspecifics over other factors. However, I don’t think the authors overstate their results in their conclusions, and I am happy that the study meets the requirements for publication in PLosOne.

I have some suggested minor revisions:

Sample sizes: Apologies if I missed it, but I can’t see any sample sizes for the different experiments? I see from Experiment 1 that there were 50 families created, but not clear if that means n = 50 for that experiment? And not sure for the other three experiments.

It might be personal preference, but I also find it useful when all data points are visible on the boxplots to give an instant idea of sample size. This is not as essential as stating the sample sizes clearly somewhere in the text though.

Throughout the manuscript, authors refer to “egg-laying” patterns or number of eggs laid, but as far as I can tell, they actually measure adult offspring (e.g. lines 264-267 imply that egg numbers are being counted). I understand that offspring is likely a good proxy for number of eggs laid, but egg-adult viability can be <100% and might differ between treatments (e.g. density on a patch can affect developmental success). I think the authors could consider using more specific language (i.e. adult offspring, not eggs), unless they did any egg counts to verify viability? Authors could add a discussion point around using number of eclosed adult offspring as a proxy for egg number.

A similar point applies to axis titles etc. “number of conspecific offspring already on patch” is not what is measured (I think). It is an assumption that the number of adult offspring that belong to the demonstrator female was equal to the number of eggs on the dish encountered by the focal female. There can always be more eggs or larvae than adult offspring.

Line 405-407: I think there is a typo here – one of the dishes contains a related female, right?

Line 358: this should refer to Figure 1a specifically, then Figure 1b should be referenced in the text somewhere (lines 363-65?) but isn’t currently.

Line 363-65: “more brown-eyed offspring eclosing from vials whose prior occupant was an unrelated female (IV-bw, mated to a IV-bwD male) than those that had previously been exposed to an unrelated female” Possible typo - should this read related?

Line 457-463: I think this sentence could be broken down into 2-3 sentences, it was quite hard to read.

Reviewer #2: Dear editor,

Please find included below my review for manuscript PONE-D-24-34027, “On the use of kinship and familiarity associated social information in mediating Drosophila melanogaster oviposition decisions”, by Emily Rakosy, Sanduni Talagala and Tristan A.F. Long. In this manuscript, the authors present a study on the use of social information on oviposition behaviour in D. melanogaster. In this species, mothers can profoundly affect the fitness of their offspring by deciding to oviposit in patches that may or may not be suitable. Determining whether a patch is suitable may be time- or resource-consuming. As a workaround, females may make use of social information about patches to decide to oviposit or not. In particular, cues left by related females may signal that a patch is particularly favourable, owing to e.g. reduced competition among developing kin and intraspecific variation for ability to use specific resources. The authors thus set out to test the hypothesis that D. melanogaster females prefer to oviposit with related females. To do so, they use a series of choice experiments in which they expose focal females to oviposition substrate previously used by genetic kin and non-kin, in addition to varying the developmental background in which these prior ovipositors develop (which might disrupt kin recognition). They find that focal females prefer to oviposit on patches previously used by related females, provided that they both grew up in a similar environment.

Overall, I think the work done is interesting, and the approach is suitable. I do, however, think that the paper can be revised to be clearer. There are various bits where the authors present information that I think would be better off elsewhere, which to me feels like things are not in the right order – though I’ll also admit this is (partially) a matter of taste.

Major comments:

Line 144-178: This feels like a disproportionate amount of text devoted to discussing the way CHCs may be used to obtain social information given that there is no direct test of how CHCs may be used by females. Admittedly, the different development backgrounds used in the experiments might help reveal such effects, but this is still a rather indirect measure. I think the length of this section can be reduced (but see next comment).

Line 154-159: This seems like an important detail for the experimental setup, but I don’t think this is sufficiently clear. Do you mean to say that kin signals are only reliable/recognizable if both individuals developed in a similar background? Or is it purely this background that is being detected, which implies kin due to oviposition sites being patchy? In the case of the latter, wouldn’t such muddying effects make “kin recognition” unreliable? This seems particularly important if there is indeed a lot of intraspecific genetic variation that can affect performance on different substrates (lines 122-125). I think this is the critical bit in this whole section (see previous comment) that I think needs to be made clearer. Reducing the amount of text devoted to the remainder of this section should free up some space to clarify this further.

Line 179-185: Could you be a bit more explicit about what you test? I know this comes in the next section / next few paragraphs on the experiments, but in its current form this feels perfunctory.

Regarding the data analysis, I have several questions/concerns.

Line 318-320: The authors imply here that they analyse the number of visits made by each female. In this case, the model setup here would be inadequate, in that this would be better modelled as a mixed-effect binomial GLM (something along the lines of visits ~ lid type + (1 | ID), where the last term describes the random effect of female ID on the probability of visits). This would actually apply to the data in the supplementary figures S2-S4. In addition, I wonder if it might be better to report the analyses for the main results first, and then these analyses that are in the supplementary material. This at least threw me off a bit when I was reading through the paper for the first time.

The data in Figures 1-4 actually show data on the eggs laid on each lid type. These are probably adequately modelled using a Poisson-based model. But looking at the type of data, I would argue that a zero-inflated Poisson model might actually be better (or alternatively, a zero-inflated negative binomial model, depending on variance in the data). It might be worth fitting some extra models to figure out which one best explains the observed data.

But looking at the data again, I wonder whether I am interpreting it correctly – are these really offspring counts, and not counts of females that have laid any eggs whatsoever on the different lid types? The numbers seem quite low for a D. melanogaster female. Or is that because the egg-laying substrate here is so small, and only supports a small number of developing offspring? It might be good to explain these low offspring counts somewhere, and perhaps even explicitly mention something like “females typically produce 3-7 offspring per lid” so people have some extra confirmation that this is actually the case.

Figure captions: All of them start with “boxplot illustrating data collected in the Xth experiment of […]”. I think it would be better to write something in the form of a conclusion as a title. Furthermore, while the description of the various boxplot elements may be informative (I think this excessive, but that is a matter of taste), I find that many of the figure captions are not very informative about what the data actually is and shows.

Minor comments:

Line 14: The abstract feels very long to me, but I’m not sure what the policy here is exactly. Regardless, I think the abstract can be written in a more direct manner.

Line 34: “did not apparently discriminate between sites differing in egg abundance” seems like it doesn’t gel with the preceding sentence saying females prefer previously-used sites over unexposed ones. Perhaps it might be better to say that while presence of eggs had a (qualitative) effect, there was no further (quantitative) effect of this prior clutch’s size on the focal female’s oviposition behaviour.

Line 38: “These results [further] highlight…” might be better, as the authors already discuss some prior work on D. melanogaster as a model for studying social behaviour.

Line 77: “see” = “but see”? There are a few other places where this happens, so perhaps not – but then I would just omit the “see” to have just the reference.

Line 121-125: The authors now bring up the benefits of co-ovipositing with kin, but this feels a bit jumbled as the text now goes back and forth on the pros and cons of co-oviposition. Perhaps it might be better to restructure the text in the introduction to first mention benefits, then costs (or vice versa). The preceding section also goes slightly back and forth on whether they are discussing the benefits of co-laying itself, as opposed to the benefits of using social information to make decisions about whether or not to co-lay. I think this material can be shuffled around to get a more cohesive story. I do realize that these things are interconnected though, so there might not be one ideal order, but it is likely worthwhile to try something different here.

Line 192-194: Can you provide details on number of generations introgressed and/or expected fraction of the genome introgressed? These results are likely in this study but ought to be mentioned here as well.

Line 196-199: Might be good to mention how many generations there have been since this divergence, instead of in years (looks to be roughly 50 generations). And while we’re at it, if you have any info on how often these have been backcrossed in that period might be nice as well.

Line 222-227: Could you add a bit to explain the point of this procedure? I.e. to obtain kin and non-kin females from shared versus unshared environments.

Line 256: “One” = “Once”?

Line 330-331: Can you be a bit clearer about this? I.e. what model setup is compared to what other model setup to test what effect? Same for other model comparisons.

Line 377-379: If this trend exists, show the data and/or a model; otherwise this can be omitted.

Line 405-407: “a dish whose demonstrator was an unrelated IV female, or a dish whose demonstrator was an unrelated IV female”; I think one of these ought to be a related IV female, right?

Line 422: “which” = “both of which”? I would think both of those are affected by degree of kinship.

Line 428: “capable to kinship discrimination” = “capable of kinship discrimination” or “able to discriminate kin”.

Figures: (1) The text on the axes is notably smaller than the labels indicating statistical tests. It might be nice to make these more similar (preferably by increasing the size of the text on the axes). (2) the order of brown- versus red-eyed individuals is reverted in Figure 1 relative to all other figures (which have brown-eyed first, then red-eyed).

Line 760 (Figure S1): I don’t think this is mentioned anywhere in the manuscript, but did you carry out any randomisation with how you configured this setup? Or is it always “related/familiar” top left, “unrelated/unfamiliar” top right, etc.? Not that I expect this to have any effect on the findings per se, but more that this is a detail that seemed to be missing.

6. PLOS authors have the option to publish the peer review history of their article (what does this mean? ). If published, this will include your full peer review and any attached files.

**Do you want your identity to be public for this peer review?** For information about this choice, including consent withdrawal, please see our Privacy Policy .

Reviewer #1: No

Reviewer #2: No

---

## [Author Response · Author response to Decision Letter 1]

4 Jan 2025

Sent: Wednesday, October 2, 2024 9:33 PM

To: Emily Rakosy <emily.rakosy@mail.utoronto.ca>

Subject: PLOS ONE Decision: Revision required [PONE-D-24-34027] - [EMID:9284e0c42c80af60]

PONE-D-24-34027

On the use of kinship and familiarity associated social information in mediating Drosophila melanogaster oviposition decisions

PLOS ONE

Dear Dr. Rakosy,

Thank you for submitting your manuscript to PLOS ONE. After careful consideration, we feel that it has merit but does not fully meet PLOS ONE’s publication criteria as it currently stands. Therefore, we invite you to submit a revised version of the manuscript that addresses the points raised during the review process.

We thank the editors and reviewers for their thoughtful feedback and comments. Below list how and where we were able to respond to these suggestions.

The reviewer's reports are included at the bottom of this email or can be accessed, together with a copy of this decision letter, by logging on to the journal editorial manager.

As you will note, the reviewers raise a number of substantial criticisms that prevent me from accepting the manuscript at this stage. Both reviewer's solicit additional clarification on certain outstanding issues including experimental rationale, sample number, data analysis, etc., and provide constructive feedback to improve the clarity and presentation of the article. Additionally, we note that the online repository link for deposited data is not working.

We look forward to receiving your revised manuscript.

Kind regards,

Madhumala Sadanandappa, PhD

Academic Editor

PLOS ONE

Journal Requirements:

We have reviewed these these style requirements, and have edited the document in the manner specified

This manuscript was not written in LaTex.

"We would like to thank Simran Mann, Lukas Ghiglione, and Harleen Taneja of the #DrosLife Lab for their fly-pushing, behavioral observations and camaraderie. Natasha B. Gallo provided helpful feed-back and constructive comments. Ruby Lindsay and Michael Steeleworthy of the Wilfrid Laurier University Library are thanked for their help with data archiving. T.A.F.L. was funded with a Natural Sciences and Engineering Research Council Discovery grants (RGPIN740 2016-06133 and RGPIN-2022-03988). This work was conducted at Wilfrid Laurier University, which exists on the traditional territory of the Neutral, Anishnawbe, and Haudenosaunee peoples."

"TAFL received support in the for of Natural Sciences and Engineering Research Council (https://www.nserc-crsng.gc.ca) Discovery grants (RGPIN-2016-06133 and RGPIN-2022-03988). The funding agency played no role in the study design, data collection and analysis, decision to publish, or preparation of the manuscript"

We have removed referenes to the funding sources in the acknowledgements. The amended statements are as follows

Acknowledgements

We would like to thank Simran Mann, Lukas Ghiglione, and Harleen Taneja of the #DrosLife Lab for their fly-pushing, behavioral observations and camaraderie. Natasha B. Gallo and two anonymous reviewers provided helpful feed-back and constructive comments. Ruby Lindsay and Michael Steeleworthy of the Wilfrid Laurier University Library are thanked for their help with data archiving. This work was conducted at Wilfrid Laurier University, which exists on the traditional territory of the Neutral, Anishnawbe, and Haudenosaunee peoples.

Funding Statement

TAFL received support in the for of Natural Sciences and Engineering Research Council (https://www.nserc-crsng.gc.ca) Discovery grants (RGPIN-2016-06133 and RGPIN-2022-03988). The funding agency played no role in the study design, data collection and analysis, decision to publish, or preparation of the manuscript.

The corresponding autor (TAFL) is affiliated with the chosen institute (Wilfrid Laurier University)

5. We notice that your supplementary figures are included in the manuscript file. Please remove them and upload them with the file type 'Supporting Information'. Please ensure that each Supporting Information file has a legend listed in the manuscript after the references list.

We have removed the supplementary figures from the manuscript, and have uploaded them as Supporting Information.

The reference list has been reviewed to ensure it is complete and correct.

Reviewers' comments:

Reviewer's Responses to Questions

Comments to the Author

1. Is the manuscript technically sound, and do the data support the conclusions?

Reviewer #1: Yes

Reviewer #2: Yes

2. Has the statistical analysis been performed appropriately and rigorously?

Reviewer #1: Yes

Reviewer #2: Yes

3. Have the authors made all data underlying the findings in their manuscript fully available?

Reviewer #1: No

Reviewer #2: Yes

4. Is the manuscript presented in an intelligible fashion and written in standard English?

Reviewer #1: Yes

Reviewer #2: Yes

5. Review Comments to the Author

Reviewer #1: I really enjoyed reading this paper – I thought it was clearly written and the results support the conclusions. The authors address the interesting question of whether relatedness of conspecific females plays a role in how Drosophila use public information to inform egg laying decisions. This is a novel study, which finds some evidence that relatedness and population of origin of conspecifics are likely important factors in female patch preferences. The authors also try to separate out some confounding factors of their main experiment. These latter assays do somewhat strengthen the hypothesis that relatedness is important, but further follow up experiments are probably required to strongly conclude that females are responding to relatedness of conspecifics over other factors. However, I don’t think the authors overstate their results in their conclusions, and I am happy that the study meets the requirements for publication in PLosOne.

We are very happy to hear that the reviewer enjoyed reading this paper, and thank them for their comments and suggetions!

I have some suggested minor revisions:

Sample sizes: Apologies if I missed it, but I can’t see any sample sizes for the different experiments? I see from Experiment 1 that there were 50 families created, but not clear if that means n = 50 for that experiment? And not sure for the other three experiments.

It might be personal preference, but I also find it useful when all data points are visible on the boxplots to give an instant idea of sample size. This is not as essential as stating the sample sizes clearly somewhere in the text though.

Thank you for the helpful suggestion. We have updated our methods section to (lines 254, 280, 299, 320) so that the sample sizes are easily available. We have also added the sample sizes to all the figure legends (Figures 1-4, S2-24) for the benefit of the reader.

Throughout the manuscript, authors refer to “egg-laying” patterns or number of eggs laid, but as far as I can tell, they actually measure adult offspring (e.g. lines 264-267 imply that egg numbers are being counted). I understand that offspring is likely a good proxy for number of eggs laid, but egg-adult viability can be <100% and might differ between treatments (e.g. density on a patch can affect developmental success). I think the authors could consider using more specific language (i.e. adult offspring, not eggs), unless they did any egg counts to verify viability? Authors could add a discussion point around using number of eclosed adult offspring as a proxy for egg number.

This is an important point. In our population there is high egg-to-adult survivorship in vials of moderate to low larval densities. We have specifically addressed this decision and our justification on Line 266-268: “We used the number of eclosed adult offspring as a proxy for the egg number laid, in this and subsequent experiments, as in the IV population at moderate larval densities, there is very high egg-to-adult survivorship (Khodaei et al. 2020).”

A similar point applies to axis titles etc. “number of conspecific offspring already on patch” is not what is measured (I think). It is an assumption that the number of adult offspring that belong to the demonstrator female was equal to the number of eggs on the dish encountered by the focal female. There can always be more eggs or larvae than adult offspring.

We have revised all the relevant figures (Figures 1, 2, 3 4, S5) to state that the y-axis depicts the number of conspeficic/focal offspring collected from the dish to better describe our measured variable and the underlying assumption.

Line 405-407: I think there is a typo here – one of the dishes contains a related female, right?

Thank you. We have corrected (this embarrassing) typo (Line 447-450: “In our fourth experiment, we set out (again) to compare the patterns of dish visitation and offspring production by focal females on three different dishes (a control/unexposed dish, a dish whose demonstrator was a related IV female, or a dish where the demonstrator was an unrelated IV female)”

Line 358: this should refer to Figure 1a specifically, then Figure 1b should be referenced in the text somewhere (lines 363-65?) but isn’t currently.

We have revised this section to specifically mention Figure 1a and 1b individually. (Lines 378 & 391)

Line 363-65: “more brown-eyed offspring eclosing from vials whose prior occupant was an unrelated female (IV-bw, mated to a IV-bwD male) than those that had previously been exposed to an unrelated female” Possible typo - should this read related?

You are correct! We have corrected this typo (Lines 393-395) “with more brown-eyed offspring eclosing from dishes whose prior occupant was a unrelated female (IV-bw, mated to a IV-bwD male; x̄=14.49; 95% CI: 13.03, 16.10) than those that had previously been exposed to a related female (IV, mated to a IV-bwD male; x̄=6.83; 95% CI: 5.95, 7.83; LR�2 =147.93, df=1, p=4.90x10-34)”

Line 457-463: I think this sentence could be broken down into 2-3 sentences, it was quite hard to read.

Thank you for the suggestion. We agree that this is a tricky passage, and have broken it up, and reworded it so that it is hopefully easier to read (Lines 507-515): “Interestingly, Malek and Long (2020) observed that a focal female’s offspring production patterns depended on the developmental environment of both the focal female and the demonstrator. In one assay, they observed focal females who developed in a protein-rich vial environment, oviposited more on dishes where the demonstrator females (that came from the same population) who had also developed on similar protein

---

## [Decision Letter · Decision Letter 1]

19 Jan 2025

PONE-D-24-34027R1On the use of kinship and familiarity associated social information in mediating Drosophila melanogaster oviposition decisionsPLOS ONE

Dear Dr. Rakosy,

Thank you for submitting your manuscript to PLOS ONE. After careful consideration, we feel that it has merit but does not fully meet PLOS ONE’s publication criteria as it currently stands. Therefore, we invite you to submit a revised version of the manuscript that addresses the points raised during the review process.

Please ensure that you address the minor comments made by the reviewers to enhance the clarity of the manuscript and correct any typographical errors. We look forward to receiving your revised manuscript.

We look forward to receiving your revised manuscript.

Kind regards,

Madhumala Sadanandappa

Academic Editor

PLOS ONE

Journal Requirements:

Reviewers' comments:

Reviewer's Responses to Questions

**Comments to the Author**

1. If the authors have adequately addressed your comments raised in a previous round of review and you feel that this manuscript is now acceptable for publication, you may indicate that here to bypass the “Comments to the Author” section, enter your conflict of interest statement in the “Confidential to Editor” section, and submit your "Accept" recommendation.

Reviewer #1: (No Response)

Reviewer #2: (No Response)

2. Is the manuscript technically sound, and do the data support the conclusions?

Reviewer #1: Yes

Reviewer #2: Yes

3. Has the statistical analysis been performed appropriately and rigorously? 

Reviewer #1: Yes

Reviewer #2: Yes

4. Have the authors made all data underlying the findings in their manuscript fully available?

Reviewer #1: Yes

Reviewer #2: Yes

5. Is the manuscript presented in an intelligible fashion and written in standard English?

Reviewer #1: Yes

Reviewer #2: Yes

6. Review Comments to the Author

Reviewer #1: Thanks to the authors for incorporating suggestions in their revised manuscript – I am happy my comments have been addressed, except this one point:

I’m afraid the edits to lines 457-463 (now 507-15…or should that be 539-48? I think many of the line numbers given in the response letter are incorrect) haven’t really helped me, and there are some sentence structure problems. Could the passage be summarised more concisely? e.g. along the lines of:

“A previous study found that D. melanogaster chose to oviposit near eggs of other females whose developmental diet matched their own, regardless of relatedness or the composition of the diet (Malek and Long, 2020).”

Reviewer #2: Dear editor,

I have read the revised version of manuscript PONE D-24-34027, “On the use of kinship and familiarity associated social information in mediating Drosophila melanogaster oviposition decisions” by Emily Rakosy, Sanduni Talagala, and Tristan A.F. Long, along with their replies to the queries raised during the first round of review. Altogether, I think the authors have substantially improved their manuscript. By and large, they have managed to either make edits that ameliorate any issues raised, or have clarified their reasoning for sticking to their original work; thank you for these efforts!

I only have some minor recommendations on e.g. typos (probably some still remain) and some phrases that I thought were unclear.

Minor comments:

Line 14: “Decisions” = Deciding”?

Line 40: I think several of these keywords can be omitted as they are also in the title (Drosophila melanogaster, oviposition).

Line 97: “… Dukas, 2009 and Battesti…” = “… Dukas, 2009, Battesti…”?

Line 164: “might not provide” = “might provide”? This doesn’t seem logical otherwise.

Line 170: “investing into mates” = “investing into mating”?

Line 439-440: “Interestingly, dishes where there had been two demonstrator females has been present were significantly…”; I don’t know what this is supposed to read exactly, but I think the “has been present” bit can be dropped.

Line 468: Period after statistical results should be a comma?

Line 520: “non kin” = “non-kin”?

Line 522: “trh” = “in the”?

Line 524: “females are selectively use” = “females selectively use”? If so, also edit “favouring” on line 525 to “favour”.

Line 527: I recommend to separate the last bit from “as demonstrator…” onward. That way you can explain how the transfer of gut bacteria etc. by females affects olfactory cues, which can be affected by other components to distort the information conferred by these cues.

Line 535-536: I had to read this a few times to get it; I think removing the commas around “and potentially misleading” might make this clearer. In fact, this sentence is a bit of a beast anyway, so maybe some general editing/splitting here might be helpful (this sentence goes all the way from line 533 to 538…).

Line 555: “rule-out” = “rule out”?

Line 574: “experiment one” = “experiment 1”? (for more consistency in style).

7. PLOS authors have the option to publish the peer review history of their article (what does this mean? ). If published, this will include your full peer review and any attached files.

**Do you want your identity to be public for this peer review?** For information about this choice, including consent withdrawal, please see our Privacy Policy .

Reviewer #1: No

Reviewer #2: No

---

## [Author Response · Author response to Decision Letter 2]

15 Feb 2025

We would like to thank the editors and referees that have taken the time to review out manuscript. Below (in bold font) we detail how we have changed the document in response to the points raised in the last review. Line numbers referenced are those in the “clean” manuscript.

View Letter

Date: Jan 19 2025 08:28PM

To: "Emily Rakosy" emily.rakosy@mail.utoronto.ca

From: "PLOS ONE" plosone@plos.org

Subject: PLOS ONE Decision: Revision required [PONE-D-24-34027R1]

PONE-D-24-34027R1

On the use of kinship and familiarity associated social information in mediating Drosophila melanogaster oviposition decisions

PLOS ONE

Dear Dr. Rakosy,

Thank you for submitting your manuscript to PLOS ONE. After careful consideration, we feel that it has merit but does not fully meet PLOS ONE’s publication criteria as it currently stands. Therefore, we invite you to submit a revised version of the manuscript that addresses the points raised during the review process.

Please ensure that you address the minor comments made by the reviewers to enhance the clarity of the manuscript and correct any typographical errors. We look forward to receiving your revised manuscript.

We look forward to receiving your revised manuscript.

Kind regards,

Madhumala Sadanandappa

Academic Editor

PLOS ONE

Journal Requirements:

Reviewers' comments:

Reviewer's Responses to Questions

Comments to the Author

1. If the authors have adequately addressed your comments raised in a previous round of review and you feel that this manuscript is now acceptable for publication, you may indicate that here to bypass the “Comments to the Author” section, enter your conflict of interest statement in the “Confidential to Editor” section, and submit your "Accept" recommendation.

Reviewer #1: (No Response)

Reviewer #2: (No Response)

2. Is the manuscript technically sound, and do the data support the conclusions?

Reviewer #1: Yes

Reviewer #2: Yes

3. Has the statistical analysis been performed appropriately and rigorously?

Reviewer #1: Yes

Reviewer #2: Yes

4. Have the authors made all data underlying the findings in their manuscript fully available?

Reviewer #1: Yes

Reviewer #2: Yes

5. Is the manuscript presented in an intelligible fashion and written in standard English?

Reviewer #1: Yes

Reviewer #2: Yes

6. Review Comments to the Author

Reviewer #1: Thanks to the authors for incorporating suggestions in their revised manuscript – I am happy my comments have been addressed, except this one point:

I’m afraid the edits to lines 457-463 (now 507-15…or should that be 539-48? I think many of the line numbers given in the response letter are incorrect) haven’t really helped me, and there are some sentence structure problems. Could the passage be summarised more concisely? e.g. along the lines of:

“A previous study found that D. melanogaster chose to oviposit near eggs of other females whose developmental diet matched their own, regardless of relatedness or the composition of the diet (Malek and Long, 2020).”

We are very pleased to read that the reviewer is (almost) completely satisfied with the changes to the document, and hope that the changes below to line 506-509 (on the clean document) is satisfactory:

A previous study found that D. melanogaster females were more likely to oviposit near eggs laid by other females whose developmental diet matched their own, regardless of their population of origin or the composition of the diet itself (Malek and Long 2020).

Reviewer #2: Dear editor,

I have read the revised version of manuscript PONE D-24-34027, “On the use of kinship and familiarity associated social information in mediating Drosophila melanogaster oviposition decisions” by Emily Rakosy, Sanduni Talagala, and Tristan A.F. Long, along with their replies to the queries raised during the first round of review. Altogether, I think the authors have substantially improved their manuscript. By and large, they have managed to either make edits that ameliorate any issues raised, or have clarified their reasoning for sticking to their original work; thank you for these efforts!

We thank you for your kind words. We are very happy with the revised manuscript: it has benefitted from the rigorous previous review, and are grateful for the opportunity to share this work with the readership of PLoS ONE.

I only have some minor recommendations on e.g. typos (probably some still remain) and some phrases that I thought were unclear.

We appreciate your eye for detail. In addition to correcting the text in the manner described below we have also sone a thorough re-reading of the manuscript to pick up any other typographical or grammatical errors.

Minor comments:

Line 14: “Decisions” = Deciding”?

The sentence has been changed to “Information produced by conspecifics can potentially be useful in making decisions as this “social information” may provide an energetically cheaper means of assessing oviposition site suitability rather than acquiring it personally.” (Lines 14-17)

Line 40: I think several of these keywords can be omitted as they are also in the title (Drosophila melanogaster, oviposition).

We have removed the redundant keywords (Lines 37-38)

Line 97: “… Dukas, 2009 and Battesti…” = “… Dukas, 2009, Battesti…”?

We have fixed the typo on line 92: “ovipositing (Sarin and Dukas 2009, Battesti et al. 2012), the presence”

Line 164: “might not provide” = “might provide”? This doesn’t seem logical otherwise.

Thank you for catching that error. The “not” has been removed (line 154)

Line 170: “investing into mates” = “investing into mating”?

We have rephrased the sentence (line 158-160) to read: “Lizé et al. (2014) described how a male fruit fly’s ability to strategically avoid mating with a sibling was impeded if they had both developed in the same environment.”

Line 439-440: “Interestingly, dishes where there had been two demonstrator females has been present were significantly…”; I don’t know what this is supposed to read exactly, but I think the “has been present” bit can be dropped.

We have rephrased this sentence (line 413-416) to read “Furthermore, dishes that had been exposed to two demonstrator females were significantly less likely to yield zero focal female offspring (8/48) than those dishes that had been exposed to either one demonstrator female (25/48) or from control dishes (32/48; �2 =13.43, df=2, p=0.0012).” which hopefully emphasizes that focal females are more likely to produce offspring when there are more conspecific (brown-eyed) eggs present – contrary to what we would have expected if they were avoiding the higher-density patches. We hope that this reads better.

Line 468: Period after statistical results should be a comma?

Thank you. This has been fixed (line 439)

Line 520: “non kin” = “non-kin”?

Thank you. This has been changed (line 489)

Line 522: “trh” = “in the”?

Thank you. This has been changed (line 491)

Line 524: “females are selectively use” = “females selectively use”? If so, also edit “favouring” on line 525 to “favour”.

We have changed the sentence (line 493-495) to read “These results suggest that ovipositing females are selective in their use of social information, and favor cues produced by their relatives, while perceived environmental familiarity does not enhance oviposition site use.”

Line 527: I recommend to separate the last bit from “as demonstrator…” onward. That way you can explain how the transfer of gut bacteria etc. by females affects olfactory cues, which can be affected by other components to distort the information conferred by these cues.

Line 535-536: I had to read this a few times to get it; I think removing the commas around “and potentially misleading” might make this clearer. In fact, this sentence is a bit of a beast anyway, so maybe some general editing/splitting here might be helpful (this sentence goes all the way from line 533 to 538…).

As suggested, we have broken up this sentence and rewritten it for clarity (lines 495-499): “Determining the specific cues being used by focal females is beyond the scope of this study as there are numerous ways that social information may be obtained. When ovipositing, female fruit flies transfer gut bacteria, CHCs, and male-specific compounds acquired during copulation to the surface of their eggs and the media (Bakula 1969, Guilhot et al. 2023, Moreira-Soto et al. 2024).”

Line 555: “rule-out” = “rule out”?

Thank you. This has been changed (line 516)

Line 574: “experiment one” = “experiment 1”? (for more consistency in style).

Thank you. This has been changed (line 533)

7. PLOS authors have the option to publish the peer review history of their article (what does this mean?). If published, this will include your full peer review and any attached files.

Do you want your identity to be public for this peer review? For information about this choice, including consent withdrawal, please see our Privacy Policy.

Reviewer #1: No

Reviewer #2: No

---

## [Editor Report · Decision Letter 2]

18 Feb 2025

On the use of kinship and familiarity associated social information in mediating Drosophila melanogaster oviposition decisions

PONE-D-24-34027R2

Dear Dr. Rakosy,

We’re pleased to inform you that your manuscript has been judged scientifically suitable for publication and will be formally accepted for publication once it meets all outstanding technical requirements.

Kind regards,

Madhumala Sadanandappa

Academic Editor

PLOS ONE

---

## [Editor Report · Acceptance letter]

PONE-D-24-34027R2

PLOS ONE

Dear Dr. Rakosy,

I'm pleased to inform you that your manuscript has been deemed suitable for publication in PLOS ONE. Congratulations! Your manuscript is now being handed over to our production team.

Kind regards,

on behalf of

Dr. Madhumala Sadanandappa

Academic Editor

PLOS ONE